# Widespread Difficulties and Applications in the Monitoring of Historical Buildings: The Case of the Realm of Venaria Reale

**Anna Bonora †, Kristian Fabbri \*,† and Marco Pretelli †**

Department of Architecture, University of Bologna, Viale Risorgimento, 40126 Bologna, Italy;
anna.bonora3@unibo.it (A.B.); marco.pretelli@unibo.it (M.P.)

\* Correspondence: kristian.fabbri@unibo.it

† These authors contributed equally to this work.

**Abstract:** Environmental monitoring represents a key step for restaurateurs to follow who strive to ensure the preservation of buildings and artifacts while allowing for people's thermal comfort. This paper describes the possibilities and main issues arising from the study of indoor microclimates. The presented case study focuses on the monitoring data analysis for two rooms of the Realm of Venaria Reale, in Turin. The adopted methodology provides for the gathering of knowledge about the history, the geometry, and the change of use in the course of the lifetime of the building. This information allows us to construct a virtual model of the building, through which it is possible to evaluate the past and present and to hypothesize future scenarios regarding the indoor environmental conditions. Moreover, this paper presents a specific index, namely the Heritage Microclimate Risk (HMR), which enables us to evaluate the risk level to which the artifacts kept within historic buildings are exposed. With that database of information, we can preemptively define which actions (managerial and structural) would need to be taken for the preservation of the artifacts and the building itself, avoiding the possible risk component taken by working on the real building.

**Keywords:** heritage microclimate risk (HMR); indoor microclimate; heritage; building simulation; conservation

## 1. Introduction

The focus of this research concerns the indoor microclimates of historical buildings, their modification, and their control through the identification and management of the parameters that characterize them. Specifically, this paper analyzes the opportunities and benefits resulting from the correct management of indoor microclimates, aimed to guarantee a preventive conservation of cultural heritage. Moreover, we will deepen our understanding of the principal issues that are typically faced during this kind of research.

Very frequently, studied artifacts and architecture need microclimatic conditions that are not standardized, and most are distinguished by particular system requirements. This underlines how the requirements of preserving cultural heritage are highly dependent on the object's or building's own history: use, as well as architectural, structural, and managerial changes over time. From this point of view, the approach proposed by Marco Pretelli and Kristian Fabbri is effective: they define the concept of Historical Indoor Microclimate (HIM) [1], which takes into account the modifications of the microclimatic characteristics of an indoor area over the years. Indeed, HIM develops a historical vision of architecture from construction to the present day. Moreover, HIM considers the relationship between those changes and the degradation of the artifacts and buildings. Once this relationship is understood, it is possible to define specific strategies aimed at preventive conservation.

## 1.1. Case Study

The Realm of Venaria Reale is situated in Italy in the province of Turin. It is one of the Sabuda Residences, part of the serial sites that UNESCO has included on the Heritage of Humanity list since 1997. The Realm is marked by the intermingling of building phases, styles, and techniques. It is the result of the intervention of six different architects: Amedeo Castellamonte, Michelangelo Garove, Filippo Juvarra, Benedetto Alfieri, Giuseppe Battista Piacenza, and Carlo Randoni. Not one of them completely realized their own vision of the building, adding to the unfinished character that distinguishes it.

The Realm of Venaria Reale and all the artifacts kept within it have undergone changes over the years, such as demolitions, architectural additions, changes in types of use, etc. These stratifications represent the keys to understanding the building and allow for the study of the evolution of different artistic and architectural techniques, as tastes and needs have changed over the years.

Today, after the biggest cultural heritage valorization and restoration project undertaken to date to ensure that "the giant became [the] Realm" [2], Venaria Reale became an important cultural and touristic attraction: exhibitions, concerts, shows, and international events are presented there. Currently, the compound is one of the most visited sites in Italy thanks to the beauty of its architecture and of the artifacts presented within it. This confirms the belief that investing in culture is as fundamental as it is strategic.

## 1.2. Scientific Literature

In the 1970s, scientist and researcher Garry Thompson (1925–2007) became interested in the study of indoor microclimates and preventive conservation, focusing some of his research on the conservation of movable assets hosted in museums [3]. In 1967, he took part in a conference [4] about climatology in museums, during which several studies concerning the analysis of microclimatic parameters (air temperature, relative humidity, lux, etc.) and their impact on heritage were presented.

In 1972, UNESCO, with the aim to ensure the best conservation of heritage and to support international cooperation, subdivided the concept of heritage into three groups: monuments, groups of buildings, and sites. For each typology, UNESCO recommends integrated protection, balancing control, and conservation.

The gradual affirmation of awareness of the significant impact that environmental conditions have on heritage conservation provoked various debates, not only about the concept of restoration and conservation but also about standards and indexes. As a result, the American Society of Heating, Refrigerating, and Air-Conditioning Engineers (ASHRAE) and other institutions defined and readjusted many times their guidelines for managing the microclimate of museums, libraries, archives, etc. [5–8].

Moreover, institutions such as the International Council of Museums (ICOM) and the International Centre for the Study of the Preservation and Restoration of Cultural Property (ICCROM) historically operate in this field, in terms of awareness and research. Different studies conducted by researchers who are affiliated with these institutions underline the importance of environmental monitoring with the aim to reduce material degradation in favor of preventive conservation. Gael De Guichez—a chemical engineer and member of ICCROM, who in the second half of the twentieth century proposed specific methodologies for museums—exemplifies this trend.

In the 1980s, indoor monitoring was expanded to the outdoors, which hosts important architectonical monuments. Other publications by Dario Camuffo [9–11] offer relevant scientific contributions to this field.

Some authors also wrote about Indoor Microclimate Quality (IMQ) [12], which is not defined by specific standards and is easily confused with Indoor Environmental Quality (IEQ) [13,14], which is defined by standards for new buildings (EN 15251).

Finally, recent research, deepening our understanding of the field of microclimates, mostly focuses on museums [15–26], evaluating models and strategies able to establish the conditions of

artifacts (e.g., evaluation protocols). Other researchers use building simulations, as we did for the study of Venaria Reale, to study the future impact of climate change [27] or to analyze hygrothermal conditions of museum storage [28]. Nevertheless, the scientific literature offers numerous case studies where several different approaches have been adopted. The standards considered are not always the same and neither are the methodologies, the monitoring instruments, software, etc. This makes the possibility of reaching a unique and common vision of the action necessary to achieve preventive conservation difficult, as well as making the replicability of a shared strategy to apply to multiple similar case studies challenging.

As mentioned in the introduction, the approach of this research follows the concept of HIM [29–31] and highlights the difficulties and the usefulness of studying the indoor microclimate for preventive conservation.

### 1.3. Study Objective

The maintenance of a building largely depends on its microclimatic conditions. We propose in this study a preventive conservation of architecture—in this case the Realm of Venaria Reale—and of artifacts having historic and artistic value through the understanding and management of the indoor microclimate.

It is clearly fundamental to identify choices and paths that allow us to preserve, conserve, and make available the cultural heritage. Unfortunately, combining the two antithetical purposes of enjoying and conserving is complicated. To do this, an approach that is both transversal and interdisciplinary is needed. The science of prevention and conservation is in fact a subject that is particularly complex, because it includes different actors from several areas, for example, from technical physics to architecture, chemistry, and environmental engineering. The aforementioned multidisciplinarity might be one of the reasons why today it is not yet completely clear what a "suitable environment" means (let alone an "environment"). Therefore, to establish the line of action necessary for a preventive conservation of cultural heritage, there is no simple or immediate solution.

The study of the indoor microclimate in the Realm of Venaria Reale presented in this paper is an opportunity to explain many reasons why there is a strong need today for a conscientious management of cultural heritage and how imperative it is considering all the difficulties in this field. Indeed, to manage a historic building, for example, we have to guarantee its sustainability, preservation, and energy efficiency, making both its preventive conservation and accessibility possible. To do this successfully, we need to know both the problems and strengths of the building and of the indoor microclimate control process.

## 2. Materials and Methods

The research methodology is structured has follows:
- Archive search and on-site visits
- Monitoring campaign
- Construction of a virtual building model
- Validation of the virtual model
- Building simulation
- Calculation of the Heritage Microclimate Risk (HMR)

It is not always necessary (or possible) to complete all these steps, and it is not even mandatory to do them in this order, but one of the strong points of the study of indoor microclimate is that all the steps of this method give us good information, allowing us to further expand our knowledge of the environmental conditions in the building and thus of the artifacts' conservation and human comfort.

### 2.1. Archive Search and On-Site Visits

This step is part of the acknowledgement of the case study and consists in finding useful material which primarily allows gathering information about the building's geometry. During this step, it is

possible to find bibliographical and archival material about the phases of construction, for example, but also of its structure, and information about the use of spaces, visitors' paths, and opening hours, etc.

In the case of the Realm of Venaria Reale, we collected much information by talking with some architects who worked on the restoration of the complex and the Centre of Conservation and Restoration (CCR) as part of the Venaria Reale team. This way, it was possible to obtain some sections, layouts, bibliographic advice, etc., for the Realm.

## 2.2. Monitoring Campaign

Currently, since the end of the restoration in 2007, there is a monitoring system inside the Realm consisting of repeaters and probes that send data wirelessly to any device.

In this case, the monitoring campaign was not performed, but the result was positive. We acquired ten years of monitoring data for the two rooms studied from the CCR, that is, from 7 August 2007 to 3 August 2011 for Room 38, and to 16 March 2017 for Room 33.

## 2.3. Construction of a Virtual Building Model and Its Validation

Having obtained the information about the geometry of the model, the outdoor climate data, the building's materials, the Heating, Ventilation and Air-Conditioning (HVAC) system operations, the opening hours, etc., we were able to produce a 3D model of the case study. For the case presented in this paper ,we made the first 3D model using the SketchUp program (version 20.0 Trimble Inc., Sunnyvale, CA, USA) and, using a plug-in, we uploaded that model to IES.VE (Virtual Environment by Integrated Environmental Solutions), a dynamic simulation software that makes it possible to elaborate information about energy use, light levels, occupant comfort, $CO_2$ emissions, airflow, etc., generating data, images, and videos.

To be validated, data produced by the virtual model in the current scenario must be compared to the real monitored data, and these have to respect the parameters reported by guideline 14 of ASHRAE. The validation parameters are as follows:

- Mean bias error (MBE):

$$MBE = \frac{\sum_{i=1}^{n}(M_i - s_i)}{\sum_{i=1}^{n} M_i} \tag{1}$$

Validated if MBE < 10%.

- Coefficient of variation root-mean-square error (RMSE):

$$CV\ (RMSE) = \frac{\sqrt{\sum_{i=1}^{n}\left[\frac{(M_i - S_i)^2}{\dot{N}_i}\right]}}{\frac{\sum_{i=1}^{n} M_i}{n}} \tag{2}$$

Validated if CV (RMSE) < 30%.

- Pearson correlation coefficient:

$$PEARSON = \frac{\sigma_{MS}}{\sigma_M \sigma_s} \tag{3}$$

If > 0.7 = strong correlation, if between 0.3 and 0.7 = correlation.

- Coefficient of determination linear regression $R^2$:

$$R^2 = \frac{\sum_{i=1}^{n}(S_i - \bar{M})^2}{\sum_{i=1}^{n}(M_i - \bar{M})^2} \tag{4}$$

Validated if $R^2$ > 0.5.

*2.4. Building Simulation*

The building simulation allows seeing the current conditions of the indoor microclimate of the building, but it is possible to hypothesize different scenarios, such as changing the building's material, or the outdoor climate, or also the doors' and windows' opening or closing. This allows verifying the consequences of several possible modifications to the building using a virtual model without the risk of damaging the actual site.

*2.5. HMR Calculation*

The Heritage Microclimate Risk is a specific index with which you can estimate the aggressiveness of the microclimate against a specific material or artifact. It is calculated as follows:

$$HMR = \frac{mr_h}{h} \tag{5}$$

and regarding the single variable (x):

$$HMR_{(x)} = \frac{mr_{h_{(x)}}}{h} \tag{6}$$

where

$mr_h$      is the hourly microclimate risk of the reference period;

h      represents the total hours of the reference period.

In the case presented in this paper, the reference period corresponds to the monitoring campaign.

The hourly microclimate risk (mrh) is determined by the following:

$$mr_h = \sum_{j=1}^{n} \left(hr_{(x)} > hr_{(x,set),min}\right)_n + \left(hr_{(x)} < hr_{(x,set),max}\right)_n \quad [h] \tag{7}$$

where

$hr_{(x)}$      is the heritage risk of the microclimatic variable (x);

$hr_{(x,set),min}$ is the heritage risk of the microclimatic variable (x) with the minimum set-point, defined as the lower range established by the Standard (UNI 10829, 1999) or other guidelines;

$hr_{(x,set),max}$ is the heritage risk of the microclimatic variable (x) with the maximum set-point defined as the lower range established by the Standard (UNI 10829, 1999) or other guidelines;

n      is the number of hours of the reference period.

## 3. Results

The two rooms studied in the Venaria Reale are in a partition of the building, next to the "Galleria Grande" on the east side. The images below (Figure 1) show on the left (a) a rendering of the Realm realized by 3D Studio Max; on the right (b) the part of the building where the two rooms are located. This last image was designed in SketchUp. The blue lines show the partition of the building, which is shown in the image on the left (a), and the red rectangles show the rooms analyzed, which is shown in the image on the right (b).

As already mentioned, it was possible to acquire much monitoring data—Temperature (T) and Relative Humidity (RH)—from the CCR, dating from 2007. The HVAC system has been operational the whole time.

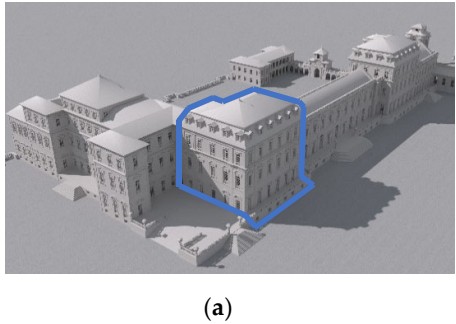
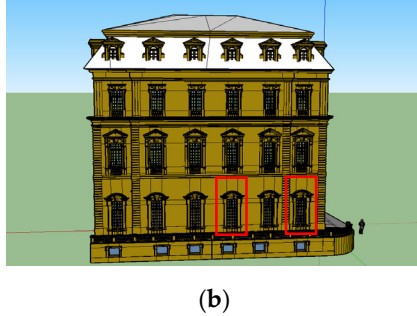

(**a**) (**b**)

**Figure 1.** 3D models of the Realm of Venaria Reale. (a) whole building with wing east side highlight; (b) front of with wing east side.

Below are some illustrative data resulting from an analysis of the probes' indoor microclimatic data. The characteristics of the current microclimate, the interpretation of the data, their relevance, and the difficulties encountered are discussed here.

The T and RH ranges in both rooms are between 15 °C and 28 °C, and 20% and 80%, respectively. The following chart (Figure 2) shows that the values of T have been recorded more frequently in Room 38. Then, if and how these values could be dangerous for the materials conservation is discussed below.

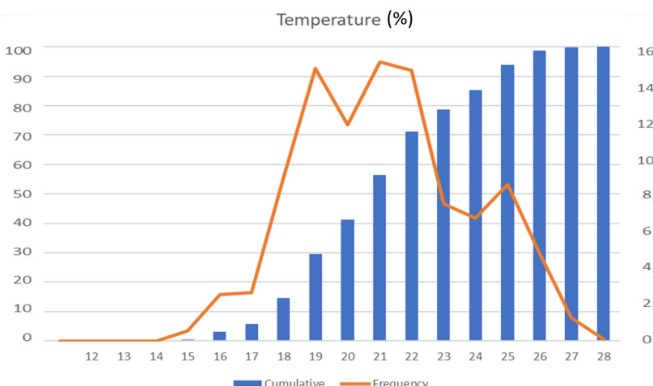

**Figure 2.** T frequency distribution and cumulated frequency, Room 38.

The frequency distribution and cumulated frequency of the indoor temperature in Room 38 provide important information, that is, the temperatures higher than 25 °C indicate a risk of overheating the materials in the room. Moreover, considering the artifacts hosted in this room (Figure 3), it is especially important to monitor the wood's conditions because it dries out more easily than other materials.

Data from both rooms include many gaps. Some are due to conscious choices made by the staff (e.g., breaks for annual periods such as from 30 August 2011 to 24 April 2013), while other shorter gaps might be caused by technical malfunctions of the probes (e.g., dead batteries or problems with the Wi-Fi connection).

As mentioned above, we acquired ten years of monitoring data for the two rooms studied from the CCR, that is, from 7 August 2007 to 3 August 2011 for Room 38, and to 16 March 2017 for Room 33. We standardized all these data and, by taking the average year by year and hour by hour, it was possible to obtain a "year-type". These data (Figure 3) are fundamental for the validation (Table 1) of the virtual environmental simulation.

**Table 1.** Temperature validation parameters, Room 33.

| MBE (%) | 0.00% | ASHRAE Guideline 14 : fail if MBE > 10% | **(Mean Bias Error)** |
|---|---|---|---|
| Count Number | 8760.00 | | |
| Sum (M-S)^2/N | 1.93 | | |
| CV (RMSE) (%) | 6% | ASHRAE Guideline 14 : fail if CV (RMSE) > 30% | |
| PEARSON | 0.78 | If >0.7 (strong) \| 0.3-0.7 (correlation) \| < 0.3 (weak) | **(Coefficient of Variation of Root Mean Square Error)** |
| MEDIA Simulated | 22.77 | | |
| R² | 0.6007 | Validated if R² > 0.5 | |

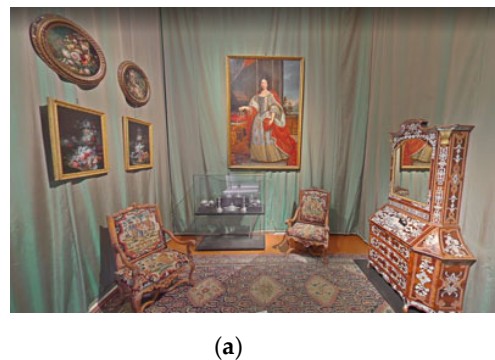 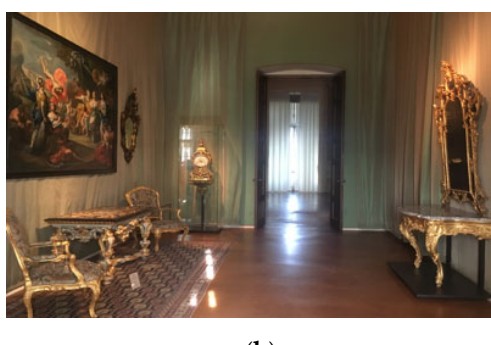

(**a**) (**b**)

**Figure 3.** Room 33, on the left (**a**); Room 38, on the right (**b**).

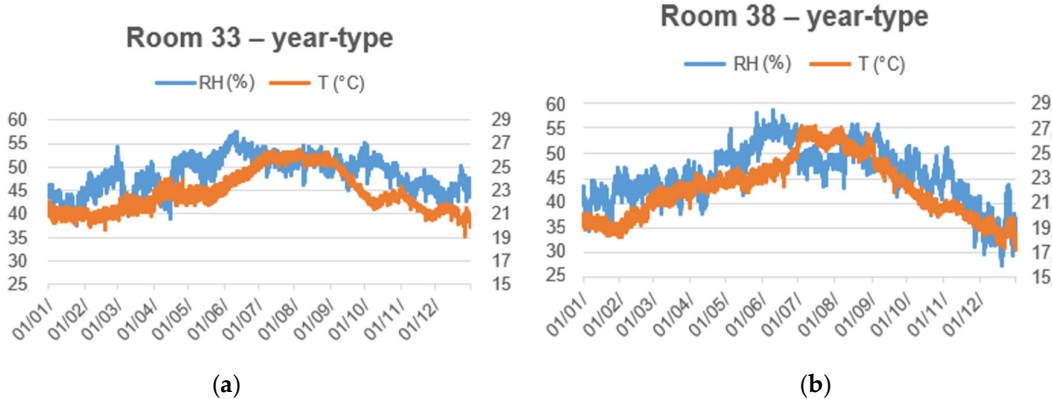

(**a**) (**b**)

**Figure 4.** year-type: T and RH in Room 33, on the left (**a**), and Room 38, on the right (**b**).

Regarding the possibility of simulating future scenarios using the virtual environmental simulation, we assumed that in 2100 the outdoor climate will rise by 1.5 °C. We simulated this hypothetical scenario to evaluate what consequences it would have on the indoor temperature and relative humidity, and therefore on the conservation of artifacts and the building.

The results show that, if the HVAC is operational, no serious modifications or damage will occur due to the indoor microclimate (Figure 5). Indeed, the mean difference between the indoor temperatures calculated on a whole year is 0.28 °C, with peaks of 0.8 °C. Considering an operational HVAC system, if there was a 1.5 °C increase in temperature, we could assume a reduction in heating costs.

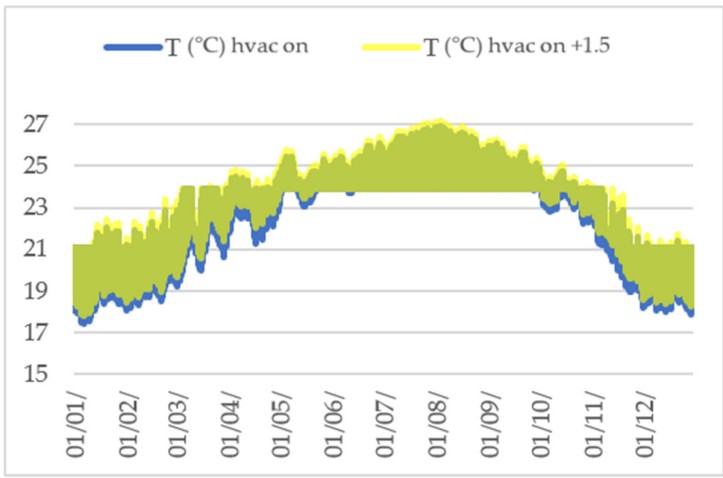

**Figure 5.** Room 33, comparative data T (°C) HVAC on.

On the contrary, if we consider that the HVAC system is switched off, we would have an annual mean difference of the indoor temperature of 1.45 °C, with a minimum peak of 0.69 °C and a maximum peak of 1.58 °C (Figure 6). We must view these data as a wake-up call—without an operational HVAC system, the effects of climate change could expose the conservation of the artifacts hosted in Venaria Reale to some risk.

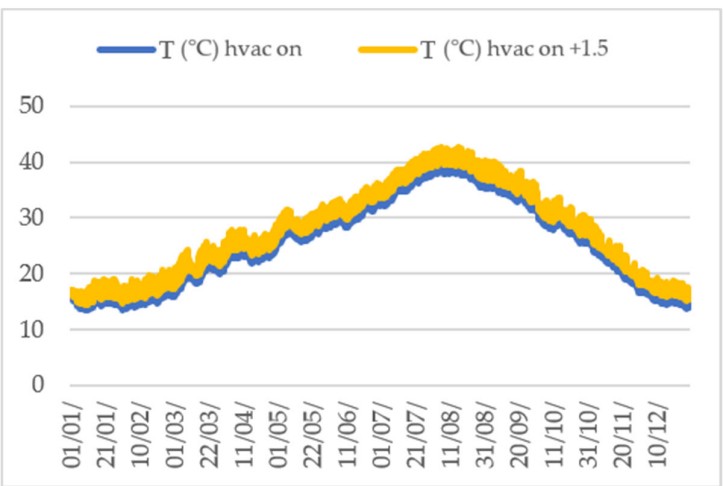

**Figure 6.** Room 33, comparative data T (°C) HVAC off.

As shown in Table 2, the maximum gap between the relative humidity recorded in 2017 and the simulated humidity in a hypothetical 2100 occurs when the HVAC system is active. If we consider the HVAC system being switched off, we see the biggest gap in the temperature between these two scenarios.

**Table 2.** Venaria Reale, T and RH comparison between 2017 and 2100.

| Room 33: Differences between 2017 and 2100 | | | | |
|---|---|---|---|---|
| | **T-HVAC ON** | **RH-HVAC ON** | **T-HVAC OFF** | **RH-HVAC OFF** |
| Min | 0.00 | 0.00 | 0.69 | 0.00 |
| Max | 0.80 | 5.72 | 1.58 | 1.83 |
| Mean | 0.28 | 2.60 | 1.45 | 0.26 |

To evaluate how the increase in T could affect the materials hosted inside the room, we calculated and compared the HMR for different materials in both scenarios, namely 2017 and 2100 (Table 3).

**Table 3.** Venaria Reale, HMR (%) of RH and T: 2017–2100, HVAC ON (on the top) and HVAC OFF (on the bottom). MIBACT [32]; UNI 10829 [33].

| | RH 2017 | RH 2100 | T 2017 | 2100 |
|---|---|---|---|---|
| **Room 33: HMR (%) - HVAC ON** | | | | |
| All materials MIBACT and UNI 10829 | 3% | 5% | 15% | 18% |
| Oil on canvas MIBACT | 29% | 25% | | |
| Oil on canvas UNI 10890 | 35% | 34% | | |
| Oil on canvas MIBACT and UNI 10829 | | | 31% | 30% |
| Wood MIBACT | 25% | 21% | | |
| Wood UNI 10890 | 26% | 23% | | |
| Wood Oil MIBACT and UNI 10829 | | | 31% | 30% |
| Fabric (silk) MIBACT | 38% | 38% | 31% | 30% |
| **Room 33: HMR (%) - HVAC OFF** | | | | |
| All materials MIBACT and UNI 10829 | 0.16% | 0.23% | 61% | 58% |
| Oil on canvas MIBACT | 15% | 15% | | |
| Oil on canvas UNI 10829 | 27% | 27% | | |
| Oil on canvas MIBACT and UNI 10829 | | | 83% | 86% |
| Wood MIBACT | 14% | 14% | | |
| Wood UN I10890 | 14% | 15% | | |
| Wood Oil MIBACT and UNI 10829 | | | 83% | 86% |
| Fabric (silk) MIBACT | 38% | 38% | 83% | 86% |

We can see that the HMR in 2100 will be more aggressive by +1% and +3%. Table 3 can be summarized as follows:

HVAC on: HMR$_T$ 2100 < 0.01 HMR$_T$ 2017; HMR$_{RH}$2100 < 0.02 HMR$_{RH}$ 2017.

HVAC off: HMR$_T$ 2100 > 0.002 HMR$_T$ 2017; HMR$_{RH}$2100 > 0.03 HMR$_{RH}$ 2017.

At this point, it would be useful to be able to evaluate the aggressiveness of the microclimate on the artifacts and the related variations due to climate change. To do that, we defined a proposal of a specific index which allows estimating a percentage level of "aggressiveness" that microclimate factors can have on materials, namely the Heritage Microclimate Risk (HMR). It is not a global or multicriterial index; indeed, it evaluates each parameter separately, taking into account specific standardized ranges per physical variable, such as material, fresco, painting, etc.

HMR is useful for making decisions about the placement or borrowing of artifacts because it allows us to verify if the conditions of a specific place—for a temporary period or definitively—are suitable or not for its conservation.

In this context, and in accordance with the aim of this paper, we must also highlight a difficulty, namely the heterogeneity of standards and indexes. Considering what is reported in UNI 10829 and MIBACT for the categories all materials, wood, oil on canvas, and fabric (silk), we can see that the range established by the standards are rather stable for the temperature parameter, whereas they vary more for the relative humidity. Indeed, the HMR value calculated considering the category "all materials" is very low. Calculating the same microclimate risk for the individual materials hosted and exhibited in Room 33, we obtain higher values. Figure 7 shows four risk envelopes. The first comprises "all materials", with a low risk. The risk then rises, varying according to the specific material ranges.

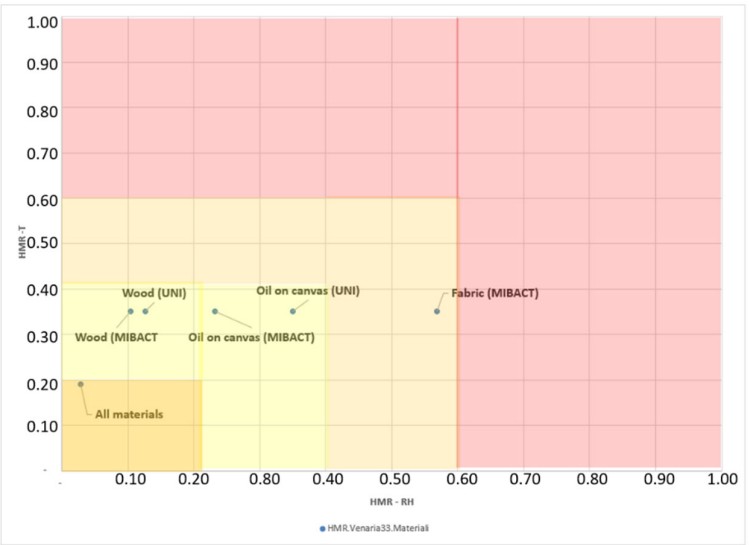

**Figure 7.** HMR Room 33 for different materials.

Moreover, there are other problems with this kind of survey in addition to the ones already listed, including the retrieval of information about the building's materials, the history of the building, and the interactions with curators and stakeholders.

## 4. Discussion

The possibility of understanding a building through the study of its indoor microclimatic conditions makes it possible to define how to avoid or minimize possible risk situations in a timely manner. This is certainly a desirable preventive approach to conservation problems, as recommendable as it is rarely practiced.

When the aim is just to obtain better conditions for the thermal comfort of visitors, making practical choices is simple, even if the concept of comfort has changed over the years. Today we do not accept that it is too cold or hot in an indoor space, and much confidence is placed in the HVAC systems. Moreover, in the last 30 to 40 years, the role of many heritage buildings has changed significantly. Often, the original use of a heritage building can change and a private home can be converted into a museum, for example. Instead of being used to host exhibitions, research, and cataloguing, a building is today increasingly used as a space that is more flexible and dynamic, keeping up with current economic, social, and tourism needs. In fact, it is not surprising today to find cafés, restaurants, and workshops inside a historical museum, and this makes the management of such historical architecture more complicated.

Therefore, when we talk about preventive conservation. we must consider three cardinal elements, establishing degrees of priority depending on the case—the building, the artifacts hosted within it, and the people who use that space. As seen in this paper, the study of the indoor microclimate associated with the virtual building simulation and the calculation of HMR makes it possible to know, in advance, the consequences that many decisions and actions could have on conservation and heritage buildings, as well as on new construction projects.

## 5. Conclusions

This paper proposes seriously considering the importance of studying indoor microclimates. It provides key information about the conditions of preserving cultural heritage. Indeed, it makes it possible to evaluate the quality of the environment for people, artifacts, and the building itself, proving to be a valid reference for restoration projects, management interventions, and prevention in historic buildings and for choices to be made about the availability to visitors of the cultural heritage.

In this study, the intangible—the air—becomes tangible, because it allows making decisions about use and conservation.

Moreover, the comparison between the indoor conditions of Rooms 33 and 38 in 2017 and 2100, assuming that climate change will cause a rise in outdoor air temperatures of 1.5 °C, shows how a building simulation associated with the study of the indoor microclimate can be used to understand hypothetical future scenarios, thereby allowing interventions in time, with the goal of putting preventive conservation in place. Therefore, understanding and managing the indoor microclimate represent a fundamental tool for architects, helping them to make timely preventive decisions.

**Author Contributions:** All the authors contributed equally to this work with regard to all statements. All authors have read and agreed to the published version of the manuscript.

**Funding:** This research received no external funding.

**Acknowledgements:** The authors would like to thank the CCR of Venaria Reale for its support with the fundamental material about the building's monitoring and the architecture.

**Conflicts of Interest:** The authors declare no conflict of interest.

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
