# Peer review of "Widespread Difficulties and Applications in the Monitoring of Historical Buildings: The Case of the Realm of Venaria Reale"

_heritage, doi:10.3390/heritage3010008_

Round 1
Reviewer 1 Report
An interesting research study and an interesting paper!
Author Response
Dear Reviewer,
many thanks for your review!
Authors
Reviewer 2 Report
The article deals with the issue of historic buildings microclimate monitoring. The authors elaborate on the concept of historical indoor microclimate and propose the index of Heritage Microclimate Risk as a potential unified measure for the successful preventive conservation of heritage buildings and movable artifacts preserved in these buildings. The application of the index is demonstrated with the case of Venaria Reale residence in Turin using the temperature and relative humidity data of two rooms of the residence. The valuable aspect of the study is the modeling of the microclimate conditions in the context of future climate change - i.e. temperature increase. The article presents valuable approach to historic building microclimate analysis and could be published after the minor revisions:
The conclusions of the paper should be more comprehensive (should present the main findings of the research) and more structured. The English language of the paper needs minor editing. The introductory section of the papers is overly extended, probably the literature review could form a separate section. Lines 62 - 71 are general description of the history of the preservation field and could be removed. Literature review should be complemented with the examples of similar research. The authors mention that the existing case studies are very diverse and have some shortcomings, so this should be demonstrated more clearly in literature review. The second section Materials and Methods presents the research methodology, however, the description in some places is linked with the case study object (for example lines 152 - 155, 160 - 164). I suggest to move the data concerning the case study object to the third Results section so that the the 2nd section would present the clear methodology and the 3rd section would show how it can be applied to the specific object. The fourth discussion section could be more elaborated subdividing it into two parts: the management proposals for the case study object deriving from the results of the analysis and the general benefits and possible applications of proposed approach.Author Response
Dear Reviewer
We report our answer to your request, in Bold and Italic.
Regards
The article deals with the issue of historic buildings microclimate monitoring. The authors elaborate on the
concept of historical indoor microclimate and propose the index of Heritage Microclimate Risk as a potential unified measure for the successful preventive conservation of heritage buildings and movable artifacts preserved in these buildings. The application of the index is demonstrated with the case of Venaria Reale residence in Turin using the temperature and relative humidity data of two rooms of the residence. The valuable aspect of the study is the modeling of the microclimate conditions in the context of future climate change - i.e. temperature increase.
The article presents valuable approach to historic building microclimate analysis and could be published after the minor revisions:
The conclusions of the paper should be more comprehensive (should present the main findings of the
research) and more structured.
#ANSWER: We added some specifications which complete the “Conclusions” section, considering that we talk about the main findings of the research in the “discussion” section too.
The English language of the paper needs minor editing.
#ANSWER: ok, an English mother tongue checked the article.
The introductory section of the papers is overly extended, probably the literature review could form a separate section.
#ANSWER: Thank you for your suggestion, following other reviewer suggestion we prefer maintain literature in Introduction paragraph.
Lines 62 - 71 are general description of the history of the preservation field and could be removed.
#ANSWER: following your suggestion we removed it.
Literature review should be complemented with the examples of similar research. The authors mention that the existing case studies are very diverse and have some shortcomings, so this should be demonstrated more clearly in literature review.
#ANSWER: Thank you, we improved literature.
The second section Materials and Methods presents the research methodology, however, the description in some places is linked with the case study object (for example lines 152 - 155, 160 - 164). I suggest to move the data concerning the case study object to the third Results section so that the the 2nd section would present the clear methodology and the 3rd section would show how it can be applied to the specific object.
#ANSWER: The descriptions (for example lines 152 - 155, 160 – 164) are specifications about the subparagraph (respectively: archive search and on-site and monitoring campaign). We took the advice, moving lines 162-164 to the 3rd section.
The fourth discussion section could be more elaborated subdividing it into two parts: the management proposals for the case study object deriving from the results of the analysis and the general benefits and possible applications of proposed approach.
#ANSWER: The HMR results of the case study presented are about a future hypothetical scenario. For this reason, in the fourth discussion section we just talked about the general benefits and possible application of the proposed approach and not about any management proposals for the case study deriving from those results about 2100 in case of a Climate Change, because it is just an hypothesis to show how the HMR calculation could be used.
Reviewer 3 Report
Dear authors,
I read your article with great interest, because i am working in the same field.
The title does not describe the contents of your paper.
The building might be described better: I missed horizontal and vertical sections of the rooms you described and the materials and properties you assumed, also from the interior.
The largest criticism I have is that you did not referenced (and read?) recent and older papers on climate change. Then you would have noticed that climate change has more effects than only increase the mean temperature: the variances become larger and it has effect also on solar radiation, relative humidity, precipitation and many other effects. in fact, your approach is much to simple. I recommend you to look at the European Project Climate for Culture: https://heritagesciencejournal.springeropen.com/articles/10.1186/s40494-015-0067-9
Your HMR index is nothing more than a percentage of time your measured (or calculated) value is within the range of accepted values. However, the exceeding value and the time of exceeding is not taken into account and might be more determining for the deterioration.
Apart from that you should describe the background of your simulation engine: you used IES.VE and I doubt if it has the possibility to take into account the hygroscopic effects of the surroundings of a room on the relative humidity inside. Furthermore, I do not read anything on the HVAC system etc. What kind of HVAC system has been used and how did you model it in IES.VE?
You described the validation by recommendations by ASHRAE, but did not describe the parameters you used to calculate MBE, CV, etc.
212: The upper range?
213: n instead of j?
247: MBE=0?
PS: Improve the English of your paper: there are a lot of mistakes in your paper.
Author Response
Dear Reviewer,
thanks. We report our answer in bold and italic.
Dear authors,
I read your article with great interest, because i am
working in the same field.
The title does not describe the contents of your paper.
#ANSWER: are happy to work at same research field! We are sorry but, actually, we cannot change the title, we’ll remind it for our future studies, thank you.
The building might be described better: I missed horizontal and vertical sections of the rooms you described and the materials and properties you assumed, also from the interior.
#ANSWER: Thank you, we are sorry but the used material is confidential, so we can’t improve the article including it : we insert the information about thickness of walls, doors, ceilings etc. and about what materials compose the building, on section “construction” on IES.VE, which calculated the respective proprieties (as conductivity; specific heat capacity; resistance; etc.).
The largest criticism I have is that you did not referenced (and read?) recent and older papers on climate change.
Then you would have noticed that climate change has more effects than only increase the mean temperature: the variances become larger and it has effect also on solar radiation, relative humidity, precipitation and many other effects. in fact, your approach is much to simple. I recommend you to look at the European Project Climate for
Culture: https://heritagesciencejournal.springeropen.com/articles/10.1186/s40494-015-0067-9
#ANSWER: Thank you. We integrated the literature with your suggestion. In this paper the hypothesis of the Climate Change is mostly the occasion to show how the simulation, associated with the HMR calculation could be used.
Your HMR index is nothing more than a percentage of time your measured (or calculated) value is within the range of accepted values. However, the exceeding value and the time of exceeding is not taken into account and might be more determining for the deterioration.
#ANSWER: We are working to find an index risk that allow to include damage.
Apart from that you should describe the background of your simulation engine: you used IES.VE and I doubt if it has the possibility to take into account the hygroscopic effects of the surroundings of a room on the relative humidity inside. Furthermore, I do not read anything on the HVAC system etc. What kind of HVAC system has been used and how did you model it in IES.VE?
#ANSWER: We considered the HVAC system and we model it in IES.VE from: toolsàbuilding template managementàthermalàI selected the ApSys as “HVAC methodology” and I specified that the variation profile has to be “ASHRAE 8am – 6pm No LunchàI set the set point of 21.1 and 23.9 in the section “space conditions”.
You described the validation by recommendations by ASHRAE, but did not describe the parameters you used to calculate MBE, CV, etc.
213: n instead of j?
247: MBE=0?
#ANSWER: We are not sur to understand your comments. We use ASHRAE guideline to do MBE, CV etc. and we calculate it from model building and monitoring data.
Line 213: changed: n instead of j.
Line 247: if we want the model to be validated, the MBE has not to be >10%, if the MBE value results <10% it is validated.
We don’t understand your comments. We use ASHRAE guideline to do MBE, CV etc. and we
PS: Improve the English of your paper: there are a lot of mistakes in your paper.
#ANSWER: ok, an English mother tongue checked the article.
Round 2
Reviewer 3 Report
The manuscript has changed to less to accept
Author Response
Dear Reviewer,
we report in attachment response and artcile marked change in red.
Regards
